# Design, Synthesis, and Bioactivity Evaluation of New Thiochromanone Derivatives Containing a Carboxamide Moiety

**DOI:** 10.3390/molecules26154391

**Published:** 2021-07-21

**Authors:** Lingling Xiao, Lu Yu, Pei Li, Jiyan Chi, Zhangfei Tang, Jie Li, Shuming Tan, Xiaodan Wang

**Affiliations:** 1School of Liquor and Food Engineering, Guizhou University, Guiyang 550025, China; an1378386891@163.com (L.X.); lyu1@gzu.edu.cn (L.Y.); qq1401064120@126.com (J.C.); tzf18885247153@163.com (Z.T.); sushilee0120@163.com (J.L.); wangxiaodan0516@126.com (X.W.); 2Qiandongnan Engineering and Technology Research Center for Comprehensive Utilization of National Medicine/Key Laboratory for Modernization of Qiandongnan Miao & Dong Medicine, Kaili University, Kaili 556011, China; 3Guizhou Provincial Key Laboratory of Fermentation Engineering and Biological Pharmacy, Guizhou University, Guiyang 550025, China

**Keywords:** thiochromanone, carboxamide, antibacterial activity, antifungal activity

## Abstract

In this study, using the botanical active component thiochromanone as the lead compound, a total of 32 new thiochromanone derivatives containing a carboxamide moiety were designed and synthesized and their in vitro antibacterial activities against *Xanthomonas oryzae* pv. *oryzae* (*Xoo*), *Xanthomonas oryzae* pv. *oryzicolaby* (*Xoc*), and *Xanthomonas axonopodis* pv. *citri* (*Xac*) were determined, as well as their in vitro antifungal activities against *Botryosphaeria dothidea* (*B. dothidea*), *Phomopsis* sp., and *Botrytis cinerea* (*B. cinerea*). Bioassay results demonstrated that some of the target compounds exhibited moderate to good in vitro antibacterial and antifungal activities. In particular, compound **4e** revealed excellent in vitro antibacterial activity against *Xoo*, *Xoc*, and *Xac*, and its EC_50_ values of 15, 19, and 23 μg/mL, respectively, were superior to those of Bismerthiazol and Thiodiazole copper. Meanwhile, compound **3b** revealed moderate in vitro antifungal activity against *B. dothidea* at 50 μg/mL, and the inhibition rate reached 88%, which was even better than that of Pyrimethanil, however, lower than that of Carbendazim. To the best of our knowledge, this is the first report on the antibacterial and antifungal activities of this series of novel thiochromanone derivatives containing a carboxamide moiety.

## 1. Introduction

Plant bacterial and fungal diseases have posed serious threats in agricultural production and in spite of the best control efforts of plant pathologists, continue to contribute to heavy crop losses worldwide each year [1,2]. In recent years, the irrational use of traditional pesticides for plant bacterial and fungal disease control have posed a danger to living systems, killing not only target bacteria and fungi, but also affecting beneficial living systems [3]. Therefore, the resistance of plant bacterial and fungal diseases against pesticides is rapidly becoming a serious problem, and in pesticide research the development of novel antibacterial and antifungal agents is still a major challenge to be tackled [4].

Chromone, a kind of botanical active component with extensive biological activities, is widely found in the secondary metabolites of flowers, roots, stems, and pericarp of many plants [5,6]. Thiochromanone, a kind of chromone compound, is an important botanical active component with extensive biological activities, including antiviral [7], antibacterial [8,9], antifungal [8,10,11,12], herbicidal [13,14], and insecticidal [15] activity. Therefore, using thiochromanone as the leading compound to develop promising agrochemical candidates will become a reality. In our previous study, we reported a series of novel thiochromanone derivatives containing a sulfonyl hydrazone moiety (Figure 1) with moderate to good antibacterial activities against *Xanthomonas oryzae* pv. *oryzae* (*Xoo*), *Xanthomonas oryzae* pv. *oryzicolaby* (*Xoc*), and *Xanthomonas axonopodis* pv. *citri* (*Xac*) [16]. Meanwhile, carboxamides, as important nitrogen-containing compounds in organic synthesis, have attracted considerable attention due to their broad range of biological activities, including antiviral [17], antibacterial [18,19], antifungal [20,21,22], herbicidal [23], and insecticidal [24,25] activity. Therefore, carboxamide could reasonably be considered as a potential active group in the design of new lead compounds.

In this study, using the botanical active component thiochromanone as the lead compound, a series of new thiochromanone derivatives containing a carboxamide moiety were designed and synthesized. We then determined the in vitro antibacterial activities of the derivatives against *Xoo*, *Xoc*, and *Xac* as well as their in vitro antifungal activities against *Botryosphaeria dothidea* (*B. dothidea*), *Phomopsis* sp., and *Botrytis cinerea* (*B. cinerea*).

## 2. Results and Discussion

### 2.1. Chemistry

The synthetic route to the target compounds **3a**–**3h** and **4a**–**4x** was carried out in three consecutive steps as shown in Scheme 1. Using a 4-substituted thiophenol as the starting material, the target compounds **3a**–**3h** and **4a**–**4x** were prepared with yields of 68–88% and their structures were determined by ^1^H NMR, ^13^C NMR, and HRMS. The ^1^H NMR, ^13^C NMR, and HRMS spectra for all the target compounds are shown in Appendix A.

In the ^1^H NMR spectra for compound **4d**, two singlets at *δ* 11.73 and 10.36 ppm indicated the presence of –OH and –NH– groups, respectively; a chemical shift at 7.85–7.14 ppm indicated the presence of hydrogen atoms of the benzene ring in the thiochromanone group; two doublet-doublets at 3.32–3.12 ppm indicated the presence of CH_2_ in the thiochromanone group. Meanwhile, in the ^13^C NMR spectra for compound **4d**, a singlet at 168.30 ppm indicated the presence of C=O in the thiochromanone group; a doublet at 159.79 and 157.41 ppm indicated the presence of C=O; a singlet at 149.88 ppm indicated the presence of C=N in the thiochromanone group.

### 2.2. Biological Evaluations

The in vitro antibacterial activities of the racemic target compounds **3a**–**3h** and **4a**–**4x** against *Xoo*, *Xoc*, and *Xac* were determined by turbidimeter tests [26,27] and the bioassay results are listed in Table 1 and Table 2. As shown in Table 1, at 200 and 100 μg/mL, some of the target compounds exhibited moderate to good antibacterial activities against *Xoo*, *Xoc*, and *Xac*. Among of them, compound **4e** at 200 μg/m, exhibited excellent in vitro antibacterial activity (100%) against *Xoo*, which was even better that that of Bismerthiazol and Thiodiazole copper. Meanwhile, as shown in Table 2, compounds **4d**, **4e**, **4f**, **4h**, and **4i** displayed in vitro antibacterial activities against *Xoo*, *Xoc*, and *Xac*, with EC_50_ values in the range of 15–29, 19–34, and 23–41 μg/mL, respectively, and their antibacterial activities were better than those of Bismerthiazol and Thiodiazole copper. In particular, compound **4d** revealed the best in vitro antibacterial activity against *Xoo*, *Xoc*, and *Xac*, and its EC_50_ values of 15, 19, and 23 μg/mL, respectively, were even better than those of Bismerthiazol and Thiodiazole copper as well as the other target compounds; however, lower than those of compound methyl 6-chloro-4-(2-((4-fluorophenyl)sulfonyl)hydrazineylidene)thiochromane-2-carboxylate [16].

Meanwhile, the in vitro antifungal activities of the racemic target compounds **3a**–**3h** and **4a**–**4x** against *B. dothidea*, *Phomopsis* sp., and *B. cinerea* were tested at 50 μg/mL by the mycelial growth rate method [28] and the results are listed in Table 3. As shown in Table 3, the target compounds revealed certain antifungal activities against *B. dothidea*, *Phomopsis* sp., and *B. cinerea* at 50 μg/mL with inhibition rate ranges of 0–22%, 0–60%, and 2–88%, respectively. In particular, compound **3b** revealed moderate antifungal activity against *B. dothidea* at 50 μg/mL, and the inhibition rate reached 88%, which was even better than that of Pyrimethanil, however, lower than that of Carbendazim.

### 2.3. Structure–Activity Relationship Analysis

The structure–activity relationship (SAR) analysis was deduced on the basis of the antibacterial and antifungal activity values listed in Table 1, Table 2 and Table 3. First, the introduction of an oxime ether or oxime fragment to the 4-position of thiochromanone can increase the antibacterial activity against *Xoo*, *Xoc*, and *Xac* (**4a** > **3a** and **4d** > **3b**); to the contrary, it can decrease the antifungal activity against *B. dothidea*, *Phomopsis* sp., and *B. cinerea* (**3a** > **4a** and **3b** > **4d**). Second, on comparing the same substituent at the R_2_ and R_3_ substituent groups, with the presence of a −Cl group at the R_1_ substituent group, the corresponding compounds presented better in vitro antibacterial and antifungal activities which followed the order **3a** (R_1_ = −Cl) > **3e** (R_1_ = −CH_3_) and **4a** (R_1_ = −Cl) > **4m** (R_1_ = −CH_3_). Third, compared with the same substituent at the R_1_ and R_3_ substituent groups, a smaller electron drawing group at the R_2_ substituent group could cause an increase in the antibacterial and antifungal activities which followed the order **3b** (R_2_ = −F) > **3c** (R_2_ = −Cl) > **3a** (R_2_ = −H) > **3d** (R_2_ = −CH_3_) and **4d** (R_2_ = −F) > **4g** (R_2_ = −Cl) > **4a** (R_2_ = −H) > **4j** (R_2_ = −CH_3_). Forth, compared with the same substituent at the R_1_ and R_2_ substituent groups, a –CH_3_ at the R_3_ substituent group could cause an increase in the antibacterial and antifungal activities which followed the order **4b** (R_3_ = −CH_3_) > **4c** (R_3_ = −C_2_H_5_) > **4a** (R_3_ = −H) and **4n** (R_3_ = −CH_3_) > **4o** (R_3_ = −C_2_H_5_) > **4m** (R_3_ = −H). 

## 3. Materials and Methods 

### 3.1. General Information

The melting points were determined by an uncorrected WRX-4 binocular microscope (Shanghai Yice Tech. Instrument Co., Shanghai, China). ^1^H NMR and ^13^C NMR spectral analyses were performed on a Bruker DRX-400 NMR spectrometer (Bruker, Rheinstetten, Germany). HRMS data were measured on a Waters Xevo G2-S QTOF mass spectrometer (Waters, Milford, MA, USA).

### 3.2. Chemical Synthesis

#### 3.2.1. Preparation Procedure of Intermediate 2

As shown in Scheme 1, intermediate **2** was prepared according to our previously reported method [16].

#### 3.2.2. Preparation Procedure for the Target Compounds **3a**–**3h**

To a 50 mL round bottom flask equipped with a magnetic stirrer, intermediate **2** (0.02 mol) was dissolved in DMF (10 mL), and then substituted phenylamine (0.02 mol), dimethylaminopyridine (DMAP, 0.0002 mol), and 1-(3-dimethylaminopropyl)-3-ethylcarbodiimide hydrochloride (EDCI, 0.03 mol) were added. The reactions were performed overnight at room temperature. Upon completion of the reaction (determined by TLC), the mixture was quenched with distilled water (50 mL) and the precipitated residues were filtered, dried under vacuum, and recrystallized from methanol to give the pure racemic target compounds **3a**–**3h**. The physical characteristics, ^1^H NMR, ^13^C NMR, and HRMS data for the target compounds **3a**–**3h** are shown below. The ^1^H NMR, ^13^C NMR, and HRMS spectra for the target compounds **3a**–**3h** are shown in Appendix A.

Data for 6-chloro-4-oxo-*N*-phenylthiochromane-2-carboxamide (**3a**). Yellow solid; mp 121–123 °C; Yield 76%; ^1^H NMR (400 MHz, DMSO-*d_6_*, ppm) *δ*: 10.33 (s, 1H, CONH), 7.88 (d, *J =* 2.4 Hz, 1H, Ph-H), 7.54–7.48 (m, 3H, Ph-H), 7.38 (d, *J =* 8.4 Hz, Ph-H), 7.28 (t, *J =* 8.0 Hz, 2H, Ph-H), 7.04 (t, *J =* 7.2 Hz, 1H, Ph-H), 4.36 (t, *J =* 4.4 Hz, 1H, SCH), 3.22 (dd, ^1^J = 4.4 Hz, ^2^J= 17.2 Hz, 1H, CH_2_), 3.13 (dd, ^1^J = 4.8 Hz, ^2^J = 16.8 Hz, 1H, CH_2_); ^13^C NMR (100 MHz, DMSO-*d_6_*, ppm) *δ*: 191.56, 169.01, 139.05, 137.29, 133.56, 132.07, 130.52, 129.77, 129.33, 127.09, 124.10, 119.48, 42.62, 40.41; HRMS (ESI) [M + Na]^+^ calcd.. for C_16_H_12_ClNO_2_S: 340.01695, found 340.01728.

Data for 6-chloro-*N*-(4-fluorophenyl)-4-oxothiochromane-2-carboxamide (**3b**). Brown solid; mp 211–213 °C; Yield 72%; ^1^H NMR (400 MHz, DMSO-*d_6_*, ppm) *δ*: 10.41 (s, 1H, CONH), 7.54–7.49 (m, 3H, Ph-H), 7.38 (d, *J =* 8.8 Hz, 1H, Ph-H), 7.16–7.09 (m, 2H, Ph-H), 4.35 (t, *J =* 4.8 Hz, 1H, SCH), 3.22 (dd, ^1^J = 4.0 Hz, ^2^J = 16.8 Hz, 1H, CH_2_), 3.13 (dd, ^1^J = 4.8 Hz, ^2^J = 16.8 Hz, 1H, CH_2_); ^13^C NMR (100 MHz, DMSO-*d_6_*, ppm) *δ*: 191.53, 168.91, 158.60 (d, *J =* 239.0 Hz), 137.24, 135.45, 135.45, 135.43, 133.57, 132.07, 130.55, 129.76, 127.11, 121.33, 121.25, 116.04, 115.81, 42.58, 40.42; HRMS (ESI) [M + Na]^+^ calcd.. for C_16_H_11_ClFNO_2_S: 358.00753, found 358.00755.

Data for 6-chloro-*N*-(4-chlorophenyl)-4-oxothiochromane-2-carboxamide (**3c**). Brown solid; mp 234–235 °C; Yield 81%; ^1^H NMR (400 MHz, DMSO-*d_6_*, ppm) *δ*: 10.43 (s, 1H, CONH), 7.88 (d, *J =* 2.4 Hz, 1H, Ph-H), 7.54–7.49 (m, 3H, Ph-H), 7.38 (d, *J =* 8.4 Hz, 1H, Ph-H), 7.16–7.11 (m, 2H, Ph-H), 4.35 (t, *J =* 4.4 Hz, 1H, SCH), 3.22 (dd, ^1^J = 4.4 Hz, ^2^J = 16.8 Hz, 1H, CH_2_), 3.13 (dd, ^1^J = 4.4 Hz, ^2^J = 16.8 Hz, 1H, CH_2_); ^13^C NMR (100 MHz, DMSO-*d_6_*, ppm) *δ*: 191.53, 168.91, 159.80, 157.41, 137.23, 135.42, 133.56, 132.06, 130.55, 129.76, 127.11, 121.32, 116.03, 42.58, 40.41; HRMS (ESI) [M + Na]^+^ calcd. for C_16_H_11_Cl_2_NO_2_S: 373.97798, found 373.98073.

Data for 6-chloro-4-oxo-*N*-(*p*-tolyl)thiochromane-2-carboxamide (**3d**). Brown solid; mp 216–218 °C; Yield 74%; ^1^H NMR (400 MHz, DMSO-*d_6_*, ppm) *δ*: 10.24 (s, 1H, CONH), 7.88 (d, *J =* 2.8 Hz, 1H, Ph-H), 7.52 (dd, ^1^J = 2.4 Hz, ^2^J = 8.4 Hz, 1H, Ph-H), 7.38 (d, *J =* 8.8 Hz, 3H, Ph-H), 7.08 (d, *J =* 8.0 Hz, 2H, Ph-H), 4.33 (t, *J =* 4.4 Hz, 1H, SCH), 3.21 (dd, ^1^J = 4.4 Hz, ^2^J = 16.8 Hz, 1H, CH_2_), 3.12 (dd, ^1^J = 4.8 Hz, ^2^J = 16.8 Hz, 1H, CH_2_), 2.23 (s, 3H, CH_3_); ^13^C NMR (100 MHz, DMSO-*d_6_*, ppm) *δ*: 191.57, 168.75, 137.35, 136.54, 133.53, 133.06, 132.09, 130.49, 129.74, 129.69, 127.08, 119.48, 40.60, 40.42, 20.88; HRMS (ESI) [M + Na]^+^ calcd. for C_17_H_14_ClNO_2_S: 354.03260, found 354.03258.

Data for 6-methyl-4-oxo-*N*-phenylthiochromane-2-carboxamide (**3e**). Yellow solid; mp 178–179 °C; Yield 75%; ^1^H NMR (400 MHz, DMSO-*d_6_*, ppm) *δ*: 10.32 (s, 1H, CONH), 7.79 (d, *J =* 0.8 Hz, 1H, Ph-H), 7.51 (dd, ^1^J = 0.8 Hz, ^2^J = 8.4 Hz, 2H, Ph-H), 7.30–7.26 (m, 3H, Ph-H), 7.20 (d, *J =* 8.0 Hz, 1H, Ph-H), 7.04 (t, *J =* 7.6 Hz, 1H, Ph-H), 4.32 (t, *J =* 4.8 Hz, 1H, SCH), 3.16 (dd, ^1^J = 4.0 Hz, ^2^J = 16.4 Hz, 1H, CH_2_), 3.09 (dd, ^1^J = 5.2 Hz, ^2^J = 16.8 Hz, 1H, CH_2_), 2.29 (s, 3H, CH_3_); ^13^C NMR (100 MHz, DMSO-*d_6_*, ppm) *δ*: 192.57, 169.25, 138.13, 135.29, 134.85, 134.78, 130.57, 129.21, 128.24, 127.59, 127.55, 121.02, 42.79, 40.98, 20.83; HRMS (ESI) [M + Na]^+^ calcd. for C_17_H_15_NO_2_S: 320.07157, found 320.07151.

Data for *N*-(4-fluorophenyl)-6-methyl-4-oxothiochromane-2-carboxamide (**3f**). Yellow solid; mp 206–207 °C; Yield 79%; ^1^H NMR (400 MHz, DMSO-*d_6_*, ppm) *δ*: 10.39 (s, 1H, CONH), 7.79 (d, *J =* 1.2 Hz, 1H, Ph-H), 7.29 (dd, ^1^J = 1.6 Hz, ^2^J = 8.0 Hz, 1H, Ph-H), 7.20 (d, *J =* 8.0 Hz, 1H, Ph-H), 7.16–7.10 (m, 2H, Ph-H), 4.31 (t, *J =* 4.4 Hz, 1H, SCH), 3.17 (dd, ^1^J = 4.4 Hz, ^2^J = 16.8 Hz, 1H, CH_2_), 3.09 (dd, ^1^J = 4.8 Hz, ^2^J = 16.8 Hz, 1H, CH_2_), 2.29 (s, 3H, CH_3_); ^13^C NMR (100 MHz, DMSO-*d_6_*, ppm) *δ*: 192.62, 168.99, 158.55 (d, *J =* 239.0 Hz), 135.58, 135.56, 135.25, 134.91, 134.83, 130.58, 128.24, 127.58, 121.25, 121.18, 115.99, 115.77, 42.79, 41.06, 20.82; HRMS (ESI) [M + Na]^+^ calcd. for C_17_H_14_FNO_2_S: 338.06215, found 338.06226.

Data for *N*-(4-chlorophenyl)-6-methyl-4-oxothiochromane-2-carboxamide (**3g**). Yellow solid; mp 209–210 °C; Yield 70%; ^1^H NMR (400 MHz, DMSO-*d_6_*, ppm) *δ*: 10.46 (s, 1H, CONH), 7.78 (s, 1H, Ph-H), 7.53 (d, *J =* 8.8 Hz, 2H, Ph-H), 7.34 (d, *J =* 8.8 Hz, 2H, Ph-H), 7.29 (d, *J =* 8.0 Hz, 1H, Ph-H), 7.20 (d, *J =* 8.0 Hz, 1H, Ph-H), 4.31 (t, *J =* 4.8 Hz, 1H, SCH), 3.16 (dd, ^1^J = 4.0 Hz, ^2^J = 16.8 Hz, 1H, CH_2_), 3.09 (dd, ^1^J = 4.8 Hz, ^2^J = 16.8 Hz, 1H, CH_2_), 2.29 (s, 3H, CH_3_); ^13^C NMR (100 MHz, DMSO-*d_6_*, ppm) *δ*: 192.57, 169.25, 138.13, 135.29, 134.85, 134.78, 130.57, 129.21, 128.24, 127.59, 121.02, 42.79, 40.98, 20.83; HRMS (ESI) [M + Na]^+^ calcd. for C_17_H_14_ClNO_2_S: 354.03260, found 354.03244. 

Data for 6-methyl-4-oxo-*N*-(*p*-tolyl)thiochromane-2-carboxamide (**3h**). Yellow solid; mp 199–200 °C; Yield 68%; ^1^H NMR (400 MHz, DMSO-*d_6_*, ppm) *δ*: 10.22 (s, 1H, CONH), 7.78 (d, *J =* 1.2 Hz, 1H, Ph-H), 7.39 (d, *J =* 8.4 Hz, 2H, Ph-H), 7.28 (dd, ^1^J = 1.6 Hz, ^2^J = 8.0 Hz, 1H, Ph-H), 7.19 (d, *J =* 8.0 Hz, 1H, Ph-H), 7.08 (d, *J =* 8.0 Hz, 2H, Ph-H), 4.29 (t, *J =* 4.8 Hz, 1H, SCH), 3.15 (d, *J =* 4.0 Hz, ^2^J = 16.8 Hz, 1H, CH_2_), 3.07 (dd, ^1^J = 4.2 Hz, ^2^J = 16.8 Hz, 1H, CH_2_), 2.29 (s, 3H, CH_3_), 2.23 (s, 3H, CH_3_); ^13^C NMR (100 MHz, DMSO-*d_6_*, ppm) *δ*: 192.66, 168.83, 136.68, 135.18, 135.03, 134.81, 132.93, 130.60, 129.65, 128.21, 127.56, 119.43, 42.82, 41.11, 20.88, 20.83; HRMS (ESI) [M + Na]^+^ calcd. for C_18_H_17_NO_2_S: 334.08722, found 334.08714.

#### 3.2.3. Preparation Procedure for the Target Compounds **4a**–**4x**

To a 50 mL round bottom flask equipped with a magnetic stirrer, a mixture of compound **3** (10 mmol), R_3_ONH_2_·HCl (15 mmol), pyridine (10 mL), and ethanol (10 mL) were added and reacted under a reflux temperature for 3–5 h. Upon completion of the reaction (determined by TLC), the mixture was cooled to room temperature and the precipitated residues were dried under vacuum and recrystallized from ethanol to give the pure racemic target compounds **4a**–**4x**. The physical characteristics, ^1^H NMR, ^13^C NMR, and HRMS data for the target compounds **4a**–**4x** are shown below. The ^1^H NMR, ^13^C NMR, and HRMS spectra for the target compounds **4a**–**4x** are shown in Appendix A.

Data for 6-chloro-4-(hydroxyimino)-*N*-phenylthiochromane-2-carboxamide (**4a**). White solid; mp 235–237 °C; Yield 80%; ^1^H NMR (400 MHz, DMSO-*d_6_*, ppm) *δ*: 11.73 (s, 1H, OH), 10.29 (s, 1H, CONH), 7.86 (t, *J =* 1.6 Hz, 1H, Ph-H), 7.54 (d, *J =* 1.2 Hz, 1H, Ph-H), 7.52 (s, 1H, Ph-H), 7.31 (d, *J =* 1.2 Hz, 2H, Ph-H), 7.29 (d, *J =* 8.0 Hz, 2H, Ph-H), 7.05 (t, *J =* 7.2 Hz, 1H, Ph-H), 4.18 (dd, ^1^J = 4.8 Hz, ^2^J = 7.2 Hz, 1H, SCH), 3.29 (dd, ^1^J = 7.2 Hz, ^2^J = 18.0 Hz, 1H, CH_2_), 3.15 (dd, ^1^J = 4.4 Hz, ^2^J = 18.0 Hz, 1H, CH_2_); ^13^C NMR (100 MHz, DMSO-*d_6_*, ppm) *δ*: 168.38, 149.92, 139.16, 132.55, 131.57, 130.63, 130.26, 129.28, 129.05, 129.45, 124.05, 119.60, 42.67, 28.32; HRMS (ESI) [M + Na]^+^ calcd. for C_16_H_13_ClN_2_O_2_S: 355.02785, found 355.02769.

Data for 6-chloro-4-(methoxyimino)-*N*-phenylthiochromane-2-carboxamide (**4b**). White solid; mp 197–198 °C; Yield 85%; ^1^H NMR (400 MHz, DMSO-*d_6_*, ppm) *δ*: 10.45 (s, 1H, CONH), 7.86 (d, *J =* 1.6 Hz, 1H, Ph-H), 7.53 (d, *J =* 8.0 Hz, 2H, Ph-H), 7.36–7.27 (m, 4H, Ph-H), 7.05 (t, *J =* 7.2 Hz, 1H, Ph-H), 4.24 (t, *J =* 5.6 Hz, 1H, SCH), 4.00 (s, 3H, CH_3_), 3.33 (dd, ^1^J = 6.4 Hz, ^2^J = 18.0 Hz, 1H, CH_2_), 3.09 (dd, ^1^J = 4.4 Hz, ^2^J = 18.0 Hz, 1H, CH_2_); ^13^C NMR (100 MHz, DMSO-*d_6_*, ppm) *δ*: 168.37, 150.63, 143.08, 139.19, 131.91, 131.39, 130.66, 130.36, 129.62, 129.25, 127.44, 124.68, 124.02, 119.57, 62.79, 40.03, 28.54; HRMS (ESI) [M + Na]^+^ calcd. for C_17_H_15_ClN_2_O_2_S: 369.04350, found 369.04279.

Data for 6-chloro-4-(ethoxyimino)-*N*-phenylthiochromane-2-carboxamide (**4c**). White solid; mp 191–192 °C; Yield 88%; ^1^H NMR (400 MHz, DMSO-*d_6_*, ppm) *δ*: 10.31 (s, 1H, CONH), 7.87 (d, *J =* 1.2 Hz, 1H, Ph-H), 7.52 (d, *J =* 7.6 Hz, 2H, Ph-H), 7.36–7.27 (m, 4H, Ph-H), 7.05 (t, *J =* 7.2 Hz, 1H, Ph-H), 4.25 (q, *J =* 7.2 Hz, 2H, CH_2_CH_3_), 4.18 (dd, ^1^J = 4.8 Hz, ^2^J = 6.8 Hz, 1H, SCH), 3.32 (dd, ^1^J = 6.8 Hz, ^2^J = 18.0 Hz, 1H, CH_2_), 3.11 (dd, ^1^J = 4.8 Hz, ^2^J = 18.0 Hz, 1H, CH_2_), 1.29 (t, *J =* 7.2 Hz, 3H, CH_2_CH_3_); ^13^C NMR (100 MHz, DMSO-*d_6_*, ppm) *δ*: 168.29, 150.34, 139.13, 131.84, 131.71, 130.71, 130.38, 129.54, 129.28, 124.70, 119.57, 70.35, 42.26, 28.78, 15.14; HRMS (ESI) [M + Na]^+^ calcd. for C_18_H_17_ClN_2_O_2_S: 383.05915, found 383.05883.

Data for 6-chloro-*N*-(4-fluorophenyl)-4-(hydroxyimino)thiochromane-2-carboxamide (**4d**). Light yellow solid; mp 225–227 °C; Yield 74%; ^1^H NMR (400 MHz, DMSO-*d_6_*, ppm) *δ*: 11.73 (s, 1H, OH), 10.36 (s, 1H, CONH), 7.85 (d, *J =* 1.2 Hz, 1H, Ph-H), 7.57–7.53 (m, 2H, Ph-H), 7.31 (d, *J =* 0.8 Hz, 2H, Ph-H), 7.14 (t, *J =* 8.8 Hz, 2H, Ph-H), 4.16 (dd, ^1^J = 4.4 Hz, ^2^J = 7.2 Hz, 1H, SCH), 3.28 (dd, ^1^J = 7.2 Hz, ^2^J = 17.6 Hz, 1H, CH_2_), 3.15 (dd, ^1^J = 4.4 Hz, ^2^J = 18.0 Hz, 1H, CH_2_); ^13^C NMR (100 MHz, DMSO-*d_6_*, ppm) *δ*: 168.30, 158.60 (d, *J =* 239.0 Hz), 149.88, 135.55, 132.55, 131.51, 130.65, 130.26, 129.19, 129.06, 124.49, 121.44, 121.36, 115.98, 115.76, 42.61, 28.31; HRMS (ESI) [M + Na]^+^ calcd. for C_16_H_12_ClFN_2_O_2_S: 373.01843, found 373.01799.

Data for 6-chloro-*N*-(4-fluorophenyl)-4-(methoxyimino)thiochromane-2-carboxamide (**4e**). White solid; mp 195–196 °C; Yield 68%; ^1^H NMR (400 MHz, DMSO-*d_6_*, ppm) *δ*: 10.35 (s, 1H, CONH), 7.86 (d, *J =* 1.6 Hz, 1H, Ph-H), 7.55–7.51 (m, 2H, Ph-H), 7.36–7.31 (m, 2H, Ph-H), 7.15–7.11 (m, 2H, Ph-H), 4.16 (dd, ^1^J = 4.8 Hz, ^2^J = 6.8 Hz, 1H, SCH), 4.00 (s, 3H, CH_3_), 3.32 (dd, ^1^J = 6.8 Hz, ^2^J = 18.0 Hz, 1H, CH_2_), 3.10 (dd, ^1^J = 4.8 Hz, ^2^J = 18.0 Hz, 1H, CH_2_); ^13^C NMR (100 MHz, DMSO-*d_6_*, ppm) *δ*: 168.22, 158.60 (d, *J =* 239.0 Hz), 150.59, 135.50, 131.79, 131.40, 130.72, 130.37, 129.64, 124.70, 121.3841, 121.34, 115.98, 115.76, 62.78, 41.99, 28.52; HRMS (ESI) [M + Na]^+^ calcd. for C_17_H_14_ClFN_2_O_2_S: 387.03408, found 387.03343.

Data for 6-chloro-4-(ethoxyimino)-*N*-(4-fluorophenyl)thiochromane-2-carboxamide (**4f**). White solid; mp 200–201 °C; Yield 78%; ^1^H NMR (400 MHz, DMSO-*d_6_*, ppm) *δ*: 10.37 (s, 1H, CONH), 7.87 (d, *J =* 1.2 Hz, 1H, Ph-H), 7.56–7.52 (m, 2H, Ph-H), 7.36–7.31 (m, 2H, Ph-H), 7.16–7.11 (m, 2H, Ph-H), 4.25 (dd, *J =* 7.2 Hz, 2H, CH_2_CH_3_), 4.17 (dd, ^1^J = 4.8 Hz, ^2^J = 7.2 Hz, 1H, SCH), 3.31 (dd, ^1^J = 7.2 Hz, ^2^J = 18.0 Hz, 1H, CH_2_), 3.12 (dd, ^1^J = 4.8 Hz, ^2^J = 18.0 Hz, 1H, CH_2_), 1.29 (t, *J =* 7.2 Hz, 3H, CH_2_CH_3_); ^13^C NMR (100 MHz, DMSO-*d_6_*, ppm) *δ*: 168.21, 158.60 (d, *J =* 239.0 Hz), 150.31, 135.53, 131.78, 131.71, 130.74, 130.78, 129.55, 124.71, 121.38 (d, *J =* 8.0 Hz), 115.98, 115.76, 70.35, 42.20, 28.77, 15.14; HRMS (ESI) [M + Na]^+^ calcd. for C_18_H_16_CLFN_2_O_2_S: 401.04973, found 401.04886.

Data for 6-chloro-*N*-(4-chlorophenyl)-4-(hydroxyimino)thiochromane-2-carboxamide (**4g**). Light yellow solid; mp 239–240 °C; Yield 79%; ^1^H NMR (400 MHz, DMSO-*d_6_*, ppm) *δ*: 11.73 (s, 1H, OH), 10.44 (s, 1H, CONH), 7.85 (s, 1H, Ph-H), 7.56 (d, *J =* 8.8 Hz, 2H, Ph-H), 7.35 (d, *J =* 8.8 Hz, 1H, Ph-H), 7.31 (d, *J =* 1.2 Hz, 3H, Ph-H), 4.17 (dd, ^1^J = 4.8 Hz, ^2^J = 7.2 Hz, 1H, SCH), 3.30 (dd, ^1^J = 6.8 Hz, ^2^J = 18.0 Hz, 1H, CH_2_), 3.13 (dd, ^1^J = 4.4 Hz, ^2^J = 18.0 Hz, 1H, CH_2_); ^13^C NMR (100 MHz, DMSO-*d_6_*, ppm) *δ*: 168.57, 149.83, 138.11, 132.56, 131.35, 130.86, 130.27 129.19, 129.06, 127.60, 124.44, 121.16, 42.51, 28.20; HRMS (ESI) [M + Na]^+^ calcd. for C_16_H_12_Cl_2_N_2_O_2_S: 388.98888, found 388.98813.

Data for 6-chloro-*N*-(4-chlorophenyl)-4-(methoxyimino)thiochromane-2-carboxamide (**4h**). Light pink solid; mp 216–218 °C; Yield 76%; ^1^H NMR (400 MHz, DMSO-*d_6_*, ppm) *δ*: 10.45 (s, 1H, CONH), 7.86 (d, *J =* 1.6 Hz, 1H, Ph-H), 7.54 (dd, ^1^J = 2.4 Hz, ^2^J = 7.2 Hz, 2H, Ph-H), 7.36–7.33 (m, 4H, Ph-H), 4.17 (dd, ^1^J = 1.6 Hz, ^2^J = 5.6 Hz, 1H, SCH), 4.00 (s, 3H, CH_3_), 3.34 (dd, ^1^J = 6.4 Hz, ^2^J = 18.0 Hz, 1H, CH_2_), 3.07 (dd, ^1^J = 4.4 Hz, ^2^J = 18.0 Hz, 1H, CH_2_); ^13^C NMR (100 MHz, DMSO-*d_6_*, ppm) *δ*: 168.50, 150.54, 138.08, 131.63, 131.40, 130.75, 129.65, 129.20, 127.60, 124.69, 121.14, 62.79, 41.90, 28.41; HRMS (ESI) [M + Na]^+^ calcd. for C_16_H_12_Cl_2_N_2_O_2_S: 403.00453, found 403.00421.

Data for 6-chloro-*N*-(4-chlorophenyl)-4-(ethoxyimino)thiochromane-2-carboxamide (**4i**). White solid; mp 200–202 °C; Yield 79%; ^1^H NMR (400 MHz, DMSO-*d_6_*, ppm) *δ*: 10.45 (s, 1H, CONH), 7.87 (d, *J =* 1.2 Hz, 1H, Ph-H), 7.55 (dd, ^1^J = 2.0 Hz, ^2^J = 6.4 Hz, 2H, Ph-H), 7.34 (dd, ^1^J = 2.0 Hz, ^2^J = 9.2 Hz, 4H, Ph-H), 4.25 (q, *J =* 7.2 Hz, 2H, CH_2_CH_3_), 4.17 (dd, ^1^J = 4.8 Hz, ^2^J = 6.8 Hz, 1H, SCH), 3.33 (dd, ^1^J = 6.8 Hz, ^2^J = 18.0 Hz, 1H, CH_2_), 3.10 (dd, ^1^J = 4.8 Hz, ^2^J = 18.0 Hz, 1H, CH_2_), 1.28 (t, *J =* 7.2 Hz, 3H, CH_2_CH_3_); ^13^C NMR (100 MHz, DMSO-*d_6_*, ppm) *δ*: 168.48, 150.26, 138.09, 131.71, 131.62, 130.77, 130.39, 129.55, 129.20, 127.60, 124.70, 121.14, 70.36, 42.11, 28.66, 15.15; HRMS (ESI) [M + Na]^+^ calcd. for C_18_H_16_Cl_2_N_2_O_2_S: 417.02018, found 417.01923.

Data for 6-chloro-4-(hydroxyimino)-*N*-(*p*-tolyl)thiochromane-2-carboxamide (**4j**). While solid; mp 249–250 °C; Yield 73%; ^1^H NMR (400 MHz, DMSO-*d_6_*, ppm) *δ*: 11.72 (s, 1H, OH), 10.20 (s, 1H, CONH), 7.85 (t, *J =* 1.6 Hz, 1H, Ph-H), 7.41 (d, *J =* 8.4 Hz, Ph-H), 7.31 (d, *J =* 1.2 Hz, 2H, Ph-H), 7.10 (d, *J =* 8.0 Hz, 2H, Ph-H), 4.15 (dd, ^1^J = 4.4 Hz, ^2^J = 7.2 Hz, 1H, SCH), 3.26 (dd, ^1^J = 7.2 Hz, ^2^J = 18.0 Hz, 1H, CH_2_), 3.14 (dd, ^1^J = 4.4 Hz, ^2^J = 18.0 Hz, 1H, CH_2_), 2.24 (s, 3H, CH_3_); ^13^C NMR (100 MHz, DMSO-*d_6_*, ppm) *δ*: 168.10, 149.94, 136.64, 133.01, 132.53, 131.67, 130.60, 129.64, 129.04, 124.45, 119.61, 42.73, 28.37, 20.90; HRMS (ESI) [M + Na]^+^ calcd. for C_17_H_15_ClN_2_O_2_S: 369.04350, found 369.04307.

Data for 6-chloro-4-(methoxyimino)-*N*-(*p*-tolyl)thiochromane-2-carboxamide (**4k**). While solid; mp 215–216 °C; Yield 79%; ^1^H NMR (400 MHz, DMSO-*d_6_*, ppm) *δ*: 10.21 (s, 1H, CONH), 7.86 (d, *J =* 1.6 Hz, 1H, Ph-H), 7.40 (d, *J =* 8.8 Hz, 2H, Ph-H), 7.36–7.30 (m, 2H, Ph-H), 7.09 (d, *J =* 8.4 Hz, 2H, Ph-H), 4.15 (dd, ^1^J = 4.8 Hz, ^2^J = 6.8 Hz, 1H, SCH), 3.98 (s, 3H, CH_3_), 3.31 (dd, ^1^J = 6.8 Hz, ^2^J = 18.0 Hz, 1H, CH_2_), 3.09 (dd, ^1^J = 4.4 Hz, ^2^J = 18.0 Hz, CH_2_), 2.23 (s, 3H, CH_3_); ^13^C NMR (100 MHz, DMSO-*d_6_*, ppm) *δ*: 168.03, 150.65, 136.62, 133.01, 131.95, 131.38, 130.66, 130.35, 129.64, 124.69, 119.58, 62.78, 42.10, 28.57, 20.89; HRMS (ESI) [M + Na]^+^ calcd. for C_18_H_17_ClN_2_O_2_S: 383.05915, found 383.05886.

Data for 6-chloro-4-(ethoxyimino)-*N*-(*p*-tolyl)thiochromane-2-carboxamide (**4l**). Light yellow solid; mp 196–198 °C; Yield 72%; ^1^H NMR (400 MHz, DMSO-*d_6_*, ppm) *δ*: 10.21 (s, 1H, CONH), 7.86 (d, *J =* 2.0 Hz, 1H, Ph-H), 7.40 (d, *J =* 8.4 Hz, 2H, Ph-H), 7.35–7.32 (m, 2H, Ph-H), 7.09 (d, *J =* 8.4 Hz, 2H, Ph-H), 4.24 (q, *J =* 6.8 Hz, 2H, CH_2_CH_3_), 4.16 (dd, ^1^J = 4.8 Hz, ^2^J = 7.2 Hz, 1H, SCH), 3.29 (dd, ^1^J = 7.2 Hz, ^2^J = 18.0 Hz, 1H, CH_2_), 3.11 (dd, ^1^J = 4.8 Hz, ^2^J = 18.0 Hz, 1H, CH_2_), 2.25 (s, 3H, CH_3_), 1.28 (t, *J =* 7.2 Hz, 3H, CH_2_CH_3_); ^13^C NMR (100 MHz, DMSO-*d_6_*, ppm) *δ*: 168.01, 150.37, 136.62, 133.01, 131.95, 131.69, 130.68, 130.36, 129.64, 129.53, 124.70, 119.59, 70.34, 42.32, 28.83, 20.90, 15.14; HRMS (ESI) [M + Na]^+^ calcd. for C_19_H_19_ClN_2_O_2_S: 397.07480, found 397.07433.

Data for 4-(hydroxyimino)-6-methyl-*N*-phenylthiochromane-2-carboxamide (**4m**). Light pink solid; mp 237–238 °C; Yield 79%; ^1^H NMR (400 MHz, DMSO-*d_6_*, ppm) *δ*: 11.46 (s, 1H, OH), 10.26 (s, 1H, CONH), 7.71 (s, 1H, Ph-H), 7.54 (d, *J =* 8.0 Hz, 2H, Ph-H), 7.30 (t, *J =* 8.0 Hz, 2H, Ph-H), 7.15 (d, *J =* 8.0 Hz, 1H, Ph-H), 7.09–7.03 (m, 2H, Ph-H), 4.10 (t, *J =* 6.4 Hz, 1H, SCH), 3.20 (d, *J =* 1.6 Hz, 2H, CH_2_), 2.28 (s, 3H, CH_3_); ^13^C NMR (100 MHz, DMSO-*d_6_*, ppm) *δ*: 168.48, 150.85, 139.22, 135.40, 130.73, 130.26, 129.42, 129.26, 128.41, 125.74, 124.00, 119.60, 43.42, 29.13, 21.19; HRMS (ESI) [M + Na]^+^ calcd. for C_17_H_16_N_2_O_2_S: 335.08247, found 335.08237.

Data for 4-(methoxyimino)-6-methyl-*N*-phenylthiochromane-2-carboxamide (**4n**). Yellow solid; mp 142–144 °C; Yield 82%; ^1^H NMR (400 MHz, DMSO-*d_6_*, ppm) *δ*: 10.26 (s, 1H, CONH), 7.71 (s, 1H, Ph-H), 7.53 (d, *J =* 7.6 Hz, 2H, Ph-H), 7.29 (t, *J =* 7.6 Hz, 2H, Ph-H), 7.17 (d, *J =* 8.0 Hz, 1H, Ph-H), 7.11 (dd, ^1^J = 1.2 Hz, ^2^J = 8.0 Hz, 1H, Ph-H), 7.05 (t, J = 7.2 Hz, 1H, Ph-H), 4.11 (dd, ^1^J = 4.8 Hz, ^2^J = 7.2 Hz, 1H, SCH), 3.96 (s, 3H, CH_3_), 3.26 (dd, ^1^J = 7.6 Hz, ^2^J = 18.0 Hz, 1H, CH_2_), 3.15 (dd, ^1^J = 4.8 Hz, ^2^J = 18.0 Hz, 1H, CH_2_), 2.28 (s, 3H, CH_3_); ^13^C NMR (100 MHz, DMSO-*d_6_*, ppm) *δ*: 168.38, 151.69, 139.21, 135.56, 130.86, 129.70, 129.63, 129.26, 128.53, 125.93, 124.00, 119.57, 62.49, 42.77, 29.37, 21.10; HRMS (ESI) [M + Na]^+^ calcd. for C_18_H_18_N_2_O_2_S: 349.09812, found 349.09763.

Data for 4-(ethoxyimino)-6-methyl-*N*-phenylthiochromane-2-carboxamide (**4o**). Yellow solid; mp 178–180 °C; Yield 80%; ^1^H NMR (400 MHz, DMSO-*d_6_*, ppm) *δ*: 10.27 (s, 1H, CONH), 7.71 (s, 1H, Ph-H), 7.53 (d, *J =* 7.6 Hz, 2H, Ph-H), 7.29 (t, *J =* 7.2 Hz, 2H, Ph-H), 7.17 (d, *J =* 8.0 Hz, 1H, Ph-H), 7.11 (dd, ^1^J = 1.6 Hz, ^2^J = 8.0 Hz, 1H, Ph-H), 7.05 (t, *J =* 7.2 Hz, 1H, Ph-H), 4.22 (q, *J =* 7.2 Hz, 2H, CH_2_CH_3_), 4.11 (dd, ^1^J = 4.8 Hz, ^2^J = 7.6 Hz, 1H, SCH), 3.24 (dd, ^1^J = 8.0 Hz, ^2^J = 18.0 Hz, 1H, CH_2_), 3.16 (dd, ^1^J = 4.8 Hz, ^2^J = 18.0 Hz, 1H, CH_2_), 2.28 (s, 3H, CH_3_), 1.28 (t, *J =* 6.8 Hz, 3H, CH_2_CH_3_); ^13^C NMR (100 MHz, DMSO-*d_6_*, ppm) *δ*: 168.37, 151.40, 139.20, 135.58, 130.77, 129.94, 129.70, 129.26, 128.55, 125.96, 124.01, 119.57, 69.99, 43.02, 29.64, 21.13, 15.19; HRMS (ESI) [M + Na]^+^ calcd. for C_19_H_20_N_2_O_2_S: 363.11377, found 363.11337.

Data for *N*-(4-fluorophenyl)-4-(hydroxyimino)-6-methylthiochromane-2-carboxamide (**4p**). White solid; mp 227–228 °C; Yield 75%; ^1^H NMR (400 MHz, DMSO-*d_6_*, ppm) *δ*: 11.46 (s, 1H, OH), 10.33 (s, 1H, CONH), 7.70 (s, 1H, Ph-H), 7.57–7.54 (m, 2H, Ph-H), 7.17–7.07 (m, 4H, Ph-H), 4.09 (t, *J =* 6.8 Hz, 1H, SCH), 3.20 (d, *J =* 6.8 Hz, 2H, CH_2_), 2.28 (s, 3H, CH_3_); ^13^C NMR (100 MHz, DMSO-*d_6_*, ppm) *δ*: 168.40, 158.58 (d, *J =* 239.0 Hz), 150.82, 135.43, 130.73, 130.27, 129.36, 128.41, 125.74, 121.43, 121.35, 115.96, 115.74, 43.33, 29.11, 21.19; HRMS (ESI) [M + Na]^+^ calcd. for C_17_H_15_FN_2_O_2_S: 353.07305, found 353.07262.

Data for *N*-(4-fluorophenyl)-4-(methoxyimino)-6-methylthiochromane-2-carboxamide (**4q**). White solid; mp 184–185 °C; Yield 70%; ^1^H NMR (400 MHz, DMSO-*d_6_*, ppm) *δ*: 10.32 (s, 1H, CONH), 7.71 (s, 1H, Ph-H), 7.56–7.52 (m, 2H, Ph-H), 7.18–7.10 (m, 4H, Ph-H), 4.09 (dd, ^1^J = 4.8 Hz, ^2^J = 7.6 Hz, 1H, SCH), 3.96 (s, 3H, CH_3_), 3.25 (dd, ^1^J = 7.6 Hz, ^2^J = 18.0 Hz, 1H, CH_2_), 3.14 (dd, ^1^J = 4.8 Hz, ^2^J = 18.0 Hz, 1H, CH_2_), 2.28 (s, 3H, CH_3_); ^13^C NMR (100 MHz, DMSO-*d_6_*, ppm) *δ*: 168.30, 158.60 (d, *J =* 239.0 Hz), 151.66, 135.60, 130.87, 129.63, 128.54, 125.94, 12140, 121.32, 115.96, 115.74, 62.50, 42.68, 29.35, 21.10; HRMS (ESI) [M + Na]^+^ calcd. for C_18_H_17_FN_2_O_2_S: 367.08870, found 367.08810.

Data for 4-(ethoxyimino)-*N*-(4-fluorophenyl)-6-methylthiochromane-2-carboxamide (**4r**). White solid; mp 168–170 °C; Yield 77%; ^1^H NMR (400 MHz, DMSO-*d_6_*, ppm) *δ*: 10.33 (s, 1H, CONH), 7.71 (s, 1H, Ph-H), 7.56–7.53 (m, 2H, Ph-H), 7.18–7.10 (m, 4H, Ph-H), 4.22 (q, *J =* 7.2 Hz, 2H, CH_2_CH_3_), 4.09 (dd, ^1^J = 4.8 Hz, ^2^J = 7.6 Hz, 1H, SCH), 3.24 (dd, ^1^J = 7.6 Hz, ^2^J = 18.0 Hz, 1H, CH_2_), 3.16 (dd, ^1^J = 4.8 Hz, ^2^J = 18.0 Hz, 1H, CH_2_), 2.28 (s, 3H, CH_3_), 1.28 (t, *J =* 7.2 Hz, 3H, CH_2_CH_3_); ^13^C NMR (100 MHz, DMSO-*d_6_*, ppm) *δ*: 168.29, 158.58 (d, *J =* 240.0 Hz), 135.61, 130.78, 129.94, 128.55, 125.96, 121.41, 121.33, 115.96, 115.74, 70.00, 42.92, 29.61, 21.12, 15.19; HRMS (ESI) [M + Na]^+^ calcd. for C_19_H_19_FN_2_O_2_S: 381.10435, found 381.10381.

Data for *N*-(4-chlorophenyl)-4-(hydroxyimino)-6-methylthiochromane-2-carboxamide (**4s**). White solid; mp 235–236 °C; Yield 78%; ^1^H NMR (400 MHz, DMSO-*d_6_*, ppm) *δ*: 11.46 (s, 1H, OH), 10.41 (s, 1H, CONH), 7.71 (s, 1H, Ph-H), 7.58–7.55 (m, 2H, Ph-H), 7.37–7.32 (m, 2H, Ph-H), 7.15 (d, *J =* 8.0 Hz, 1H, Ph-H), 7.08 (dd, ^1^J = 2.0 Hz, ^2^J = 8.0 Hz, 1H, Ph-H), 4.01 (dd, ^1^J = 5.6 Hz, ^2^J = 7.2 Hz, 1H, SCH), 3.22 (dd, ^1^J = 7.2 Hz, ^2^J = 18.0 Hz, 1H, CH_2_), 3.17 (dd, ^1^J = 5.2 Hz, ^2^J = 18.0 Hz, 1H, CH_2_), 2.28 (s, 3H, CH_3_); ^13^C NMR (100 MHz, DMSO-*d_6_*, ppm) *δ*: 168.68, 150.77, 138.18, 135.45, 130.75, 130.27, 129.18, 128.41, 127.55, 125.73, 121.16, 43.22, 29.00, 21.19; HRMS (ESI) [M + Na]^+^ calcd. for C_17_H_15_ClN_2_O_2_S: 369.04350, found 369.04330.

Data for *N*-(4-chlorophenyl)-4-(methoxyimino)-6-methylthiochromane-2-carboxamide (**4t**). White solid; mp 197–199 °C; Yield 70%; ^1^H NMR (400 MHz, DMSO-*d_6_*, ppm) *δ*: 10.41 (s, 1H, CONH), 7.71 (s, 1H, Ph-H), 7.55 (dd, ^1^J = 2.0 Hz, ^2^J = 6.8 Hz, 2H, Ph-H), 7.34 (dd, ^1^J = 2.0 Hz, ^2^J = 6.4 Hz, 2H, Ph-H), 7.16 (d, *J =* 8.0 Hz, 1H, Ph-H), 7.11 (dd, ^1^J = 1.2 Hz, ^2^J = 8.0 Hz, 1H, Ph-H), 4.11 (dd, ^1^J = 4.8 Hz, ^2^J = 7.2 Hz, 1H, SCH), 3.96 (s, 3H, CH_3_), 3.27 (dd, ^1^J = 7.2 Hz, ^2^J = 18.0 Hz, 1H, CH_2_), 3.13 (dd, ^1^J = 4.8 Hz, ^2^J = 18.0 Hz, 1H, CH_2_), 2.28 (s, 3H, CH_3_); ^13^C NMR (100 MHz, DMSO-*d_6_*, ppm) *δ*: 168.59, 151.61, 138.16, 135.62, 130.88, 129.64, 129.46, 129.17, 128.53, 127.54, 125.92, 121.13, 62.50, 42.58, 29.23, 21.10; HRMS (ESI) [M + Na]^+^ calcd. for C_18_H_17_ClN_2_O_2_S: 383.05915, found 383.05863. 

Data for *N*-(4-chlorophenyl)-4-(ethoxyimino)-6-methylthiochromane-2-carboxamide (**4u**). White solid; mp 169–170 °C; Yield 79%; ^1^H NMR (400 MHz, DMSO-*d_6_*, ppm) *δ*: 10.41 (s, 1H, CONH), 7.71 (s, 1H, Ph-H), 7.56 (dd, ^1^J = 2.0 Hz, ^2^J = 7.2 Hz, 2H, Ph-H), 7.36 (d, *J =* 3.2 Hz, 1H, Ph-H), 7.34 (d, *J =* 2.0 Hz, 1H, Ph-H), 7.17 (d, *J =* 8.0 Hz, 1H, Ph-H), 7.10 (d, *J =* 8.4 Hz, 1H, Ph-H), 4.22 (q, *J =* 7.2 Hz, 2H, CH_2_CH_3_), 4.10 (dd, ^1^J = 4.8 Hz, ^2^J = 7.6 Hz, 1H, SCH), 3.26 (dd, ^1^J = 7.6 Hz, ^2^J = 18.0 Hz, 1H, CH_2_), 3.15 (dd, ^1^J = 4.8 Hz, ^2^J = 18.0 Hz, 1H, CH_2_), 2.28 (s, 3H, CH_3_), 1.28 (t, *J =* 7.2 Hz, 3H, CH_2_CH_3_); ^13^C NMR (100 MHz, DMSO-*d_6_*, ppm) *δ*: 168.57, 151.31, 138.16, 135.63, 130.78, 129.95, 129.46, 129.18, 128.54, 127.55, 125.95, 121.14, 70.00, 42.81, 29.49, 21.13, 15.19; HRMS (ESI) [M + Na]^+^ calcd. for C_19_H_19_ClN_2_O_2_S: 397.07480, found 397.07421. 

Data for 4-(hydroxyimino)-6-methyl-*N*-(*p*-tolyl)thiochromane-2-carboxamide (**4v**). White solid; mp 237–239 °C; Yield 76%; ^1^H NMR (400 MHz, DMSO-*d_6_*, ppm) *δ*: 11.45 (s, 1H, OH), 10.17 (s, 1H, CONH), 7.70 (s, 1H, Ph-H), 7.42 (d, *J =* 8.4 Hz, 2H, Ph-H), 7.17–7.07 (m, 3H, Ph-H), 7.15 (d, *J =* 8.0 Hz, 1H, Ph-H), 7.17–7.07 (m, 4H, Ph-H), 4.08 (dd, ^1^J = 5.6 Hz, ^2^J = 7.2 Hz, 1H, SCH), 3.22 (dd, ^1^J = 5.6 Hz, ^2^J = 18.4 Hz, 1H, CH_2_), 3.16 (dd, ^1^J = 8.0 Hz, ^2^J = 18.0 Hz, 1H, CH_2_), 2.27 (s, 3H, CH_3_), 2.24 (s, 3H, CH_3_); ^13^C NMR (100 MHz, DMSO-*d_6_*, ppm) *δ*: 168.20, 150.88, 136.71, 135.37, 132.95, 130.72, 130.25, 129.62, 128.40, 125.74, 119.61, 43.50, 29.20, 21.19, 20.90; HRMS (ESI) [M + Na]^+^ calcd. for C_18_H_18_N_2_O_2_S: 349.09812, found 349.09779. 

Data for 4-(methoxyimino)-6-methyl-*N*-(*p*-tolyl)thiochromane-2-carboxamide (**4w**). White solid; mp 205–207 °C; Yield 73%; ^1^H NMR (400 MHz, DMSO-*d_6_*, ppm) *δ*: 10.17 (s, 1H, CONH), 7.71 (s, 1H, Ph-H), 7.41 (d, *J =* 8.4 Hz, 2H, Ph-H), 7.16 (d, *J =* 8.0 Hz, 1H, Ph-H), 7.12 (d, *J =* 1.6 Hz, 1H, Ph-H), 7.09 (d, *J =* 8.0 Hz, 2H, Ph-H), 4.09 (dd, ^1^J = 5.2 Hz, ^2^J = 8.0 Hz, 1H, SCH), 3.96 (s, 3H, CH_3_), 3.24 (dd, ^1^J = 7.6 Hz, ^2^J = 18.0 Hz, 1H, CH_2_), 3.14 (dd, ^1^J = 4.8 Hz, ^2^J = 18.0 Hz, 1H, CH_2_), 2.28 (s, 3H, CH_3_), 2.24 (s, 3H, CH_3_); ^13^C NMR (100 MHz, DMSO-*d_6_*, ppm) *δ*: 168.10, 151.72, 136.69, 135.53, 132.95, 130.86, 129.81, 129.62, 128.52, 125.94, 119.59, 62.49, 42.85, 29.43, 21.10, 20.90; HRMS (ESI) [M + Na]^+^ calcd. for C_19_H_20_N_2_O_2_S: 363.11377, found 363.11303.

Data for 4-(ethoxyimino)-6-methyl-*N*-(*p*-tolyl)thiochromane-2-carboxamide (**4x**). White solid; mp 198–200 °C; Yield 78%; ^1^H NMR (400 MHz, DMSO-*d_6_*, ppm) *δ*: 10.18 (s, 1H, CONH), 7.71 (s, 1H, Ph-H), 7.42 (d, *J =* 8.4 Hz, 2H, Ph-H), 7.16 (d, *J =* 8.0 Hz, 1H, Ph-H), 7.11 (d, *J =* 1.6 Hz, 1H, Ph-H), 7.09 (d, *J =* 8.4 Hz, 2H, Ph-H), 4.22 (q, *J =* 6.8 Hz, 2H, CH_2_CH_3_), 4.09 (dd, ^1^J = 5.2 Hz, ^2^J = 7.6 Hz, 1H, SCH), 3.23 (dd, ^1^J = 8.0 Hz, ^2^J = 18.4 Hz, 1H, CH_2_), 3.16 (dd, ^1^J = 5.2 Hz, 2J= 18.0 Hz, 1H, CH_2_), 2.28 (s, 3H, CH_3_), 2.24 (s, 3H, CH_3_), 1.28 (t, *J =* 7.2 Hz, 3H, CH_2_CH_3_); ^13^C NMR (100 MHz, DMSO-*d_6_*, ppm) *δ*: 168.08, 151.43, 136.69, 135.55, 132.96, 130.76, 129.92, 129.82, 129.62, 128.53, 125.96, 119.59, 69.99, 43.11, 29.70, 21.12, 20.90, 15.19; HRMS (ESI) [M + Na]^+^ calcd. for C_20_H_22_N_2_O_2_S: 377.12942, found 377.12911.

### 3.3. Bioactivity Evaluation

#### 3.3.1. Bacterial and Fungal Strains

All bacteria used in this study were provided by Guizhou University and all fungal strains used in this study were provided by Guiyang University.

#### 3.3.2. In Vitro Antibacterial Activity Test

Each target compound (7.5 mg) was dissolved in 150 μL DMSO and then 80 and 40 μL of the solution, respectively, was poured into two 15 mL centrifuge tubes each containing 4 mL 0.1% Twain aqueous solution. The solutions (1 mL) were then added into glass test tubes each containing 4 mL nutrient broth (NB) medium to prepare 5 mL test solutions with concentrations of 200 and 100 μg/mL, respectively. Then, 40 μL of the NB mediums containing *Xoo*, *Xoc*, and *Xac*, respectively, were added to the test tubes mentioned above. The inoculated test tubes were incubated at 30 °C and 180 rpm for 24–48 h until the OD_595_ values of the negative control reached 0.6–0.8 (the logarithmic growth phase). DMSO served as the negative control, whereas Thiodiazole copper and Bismerthiazol served as positive controls. Three replicates were conducted for each treatment. The OD_595_ values of the cultures were monitored on a Multiskan Sky 1530 spectrophotometer (Thermo Scientific, Poland). The inhibition rate *I* (%) was calculated by the following formula (1), where C is the corrected turbidity value of the untreated NB medium, and T is the corrected turbidity value of the treated NB medium.
Inhibition rate *I* (%) = (C–T)/C × 100(1)

On the basis of the preliminary bioassay results, the antibacterial activities (expressed by EC_50_) of some of the target compounds against *Xoo*, *Xoc* and *Xac* were evaluated and calculated using SPSS 17.0 software. Three replicates were conducted for each treatment.

#### 3.3.3. In Vitro Antifungal Activity Test

Each target compound (5 mg) was dissolved in 1 mL DMSO and mixed with 90 mL potato dextrose agar (PDA) medium. The mixed PDA mediums were then poured into 6 dishes and cooled to room temperature to prepare the PDA plates with the test solution concentration of 50 μg/mL. Mycelia dishes of approximately 0.4 cm diameter were then cut from the culture medium and picked up with a germfree inoculation needle and placed into the middle of PDA plates aseptically. The inoculated PDA plates were fostered in an incubator at 28 °C for 3–4 days until the colony diameter of the negative control reached 5–6 cm. DMSO served as the negative control, whereas Pyrimethanil and Carbendazim acted as positive controls. Three replicates were conducted for each treatment. The inhibition rate *I* (%) was calculated by the following formula (2), where C (cm) represents the diameter of fungi growth on the untreated PDA plate, and T (cm) represents the diameter of fungi on the treated PDA plate.
Inhibition rate *I* (%) = [(C−T)/(C−0.4)] × 100(2)

## 4. Conclusions

In this study, a total of 32 new thiochromanone derivatives containing a carboxamide moiety were designed and synthesized. The bioassay results demonstrated that compound **4e** exhibited excellent in vitro antibacterial activity against *Xoo*, *Xoc*, and *Xac* which was superior to those of Bismerthiazol and Thiodiazole copper. Meanwhile, compound **3b** revealed moderate in vitro antifungal activity against *B. dothidea* at 50 μg/mL which was even better than that of Pyrimethanil, nevertheless, lower than that of Carbendazim. For controlling plant bacterial and fungal diseases, this study provided a practical tool for guiding the design and synthesis of novel and more promising active small molecules of thiochromanone derivatives containing a carboxamide moiety. 

## Data Availability

Data present in this study are available on request from the corresponding author.

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
