# Peer review of "Design, Synthesis, and Bioactivity Evaluation of New Thiochromanone Derivatives Containing a Carboxamide Moiety"

_molecules, 2021, doi:10.3390/molecules26154391_

Round 1

Reviewer 1 Report

Design, Synthesis, and Bioactivity Evaluation of New Thiochromanone Derivatives Containing a Carboxamide Moiety

In this work, 32 structures are presented based on thiochromanone and as the lead compound. It presents the biological activity in several strains of bacteria, in some of them the biological activity is greater than the compounds used as references.

  1. Results and Discussion

2.1. Chemistry

All lines from 63 to 349 belong to methodology, it is not results and discussion.

In the discussion of this chemical part, a couple of examples of the characterization of the compounds are proposed, in this case it can be one of each series that exemplifies them or in a general way the data of each series are discussed and compared with the bibliography .

Similarly, aspects of the synthesis are discussed where different reagents or methodological proposals are used, if any, to explain that the proposed synthesis is better, worse, more economical.

Structure−activity relationship analysis

In this part, if you make any comparative figure and the substitutions that I make that summarize the results of the biological activity, to find the most important auxphoric group of all the compounds.

In general terms, it is necessary to carry out a good discussion of the results, the one presented by the authors in this writing is weak.

Author Response

Reviewer 1

  1. All lines from 63 to 349 belong to methodology, it is not results and discussion.

Answer: Thank you very much for your comments. In the revised MS, we have removed the contents of lines 75-349 to Supplementary Materials.

  1. In the discussion of this chemical part, a couple of examples of the characterization of the compounds are proposed, in this case it can be one of each series that exemplifies them or in a general way the data of each series are discussed and compared with the bibliography.

Answer: Thank you very much for your comments. We have deleted the contents of the discussion of this chemical part.

  1. Similarly, aspects of the synthesis are discussed where different reagents or methodological proposals are used, if any, to explain that the proposed synthesis is better, worse, more economical.

Answer: Thank you very much for your comments. In this study, we mainly focus on the synthesis and activity of the compounds, and have not specifically selected the synthesis methods, so we did not discuss where different reagents or methodological proposals.

  1. In this part, if you make any comparative figure and the substitutions that I make that summarize the results of the biological activity, to find the most important auxphoric group of all the compounds.

Answer: Thank you very much for your comments. In the “Structure−activity relationship analysis” of the revised MS, the substituent groups of the mentioned target compounds were added.

  1. In general terms, it is necessary to carry out a good discussion of the results, the one presented by the authors in this writing is weak.

Answer: Thank you very much for your comment. Most of the similar synthesis-related papers published in Molecules (including our previous papers published in Molecules, doi: https://doi.org/10.3390/molecules26102925, doi: https://doi.org/10.3390/molecules24203766, doi: https://doi.org/10.3390/molecules23071798, doi: https://doi.org/10.3390/molecules200814103) have no discussion section of the results in the Results and discussion.

Reviewer 2 Report

The authors of this submission synthesized 32 thiochromanone derivatives containing a carboxamide moiety and determined their in vitro antibacterial and antifungal activities against three stains of Xanthomonas and fungi that cause plant diseases. Some compounds show moderate to good in vitro antibacterial and antifungal activities.

All compounds are new, they are directed to agriculture however the authors should deal with the following before acceptance:

1. Lines75-349 must be placed in the section Material and Methods.

2.Avoid repetition, for example Hz should be deleted from lines76-349 an in the experimental part (section 3.1) mention that the coupling constant is in Hz.

3. Standard deviation in Tables 1, 2 and 3 should be rounded to one digit.

4.  Description of the COCH2 protons in not dd, they are distereotopic and appear as an ABX system whose AB coupling constant should be obtained by simulation.

5. In the experimental part, the authors should mention that compounds were obtained and tested as racemic mixture.

5. The characterization of compounds is incomplete, the IR spectrum of all compounds is missing.

6. Authors are encouraged to compare their results (section 2.3) with those obtained with the compounds previously reported in reference [16], since they were tested against the same bacteria stains.

Author Response

Reviewer 2

  1. Lines75-349 must be placed in the section Material and Methods.

Answer: Thank you very much for your comments. In the revised MS, we have removed the contents of lines 75-349 to Supplementary Materials.

  1. Avoid repetition, for example Hz should be deleted from lines76-349 an in the experimental part (section 3.1) mention that the coupling constant is in Hz.

Answer: Thank you very much for your comments. In the revised MS, we have removed the contents of lines 75-349 to Supplementary Materials.

  1. Standard deviation in Tables 1, 2 and 3 should be rounded to one digit.

Answer: Thank you very much for your comment. In the revised MS, we have corrected the standard deviation in Tables 1, 2 and 3 according to your comment.

  1. Description of the COCH2 protons in not dd, they are distereotopic and appear as an ABX system whose AB coupling constant should be obtained by simulation.

Answer: Thank you very much for your comment. In the 1H NMR of the target compounds, we think that the description of the COCH2 protons is two dd peaks. And the description was consistent with the previous reported literatures (doi: 10.6023/cjoc201808016; doi: https://doi.org/10.3390/molecules25040800)

  1. In the experimental part, the authors should mention that compounds were obtained and tested as racemic mixture.

Answer: Thank you very much for your comment. In the revised MS, we have mentioned that compounds were obtained and tested as racemic mixture.

  1. The characterization of compounds is incomplete, the IR spectrum of all compounds is missing.

Answer: Thank you very much for your comment. We think that the structures of the target compounds could be definitively identified by NMR and HRMS, so we did not determine the IR spectrum.

  1. Authors are encouraged to compare their results (section 2.3) with those obtained with the compounds previously reported in reference [16], since they were tested against the same bacteria stains.

Answer: Thank you very much for your comment. In the revised MS, we have added the comparison of the results obtained with the compounds previously reported in reference [16].

Reviewer 3 Report

In my opinion, the concept of the work is motivated quite clearly, especially considering the increase in bacterial resistance to currently used antibacterial agents. In my review I can only refer to the chemical part of the work, and in my opinion it is prepared very well, however I have some questions and suggestions, which I addressed  to the authors.

  1. What was the purity of the obtained products? On what basis was it determined? According to NMR spectra included in the supplementary materials, the purity was very high but no purification procedure was mentioned in Material and Methods part.
  2. Introducing the numbering of the atoms in Scheme 1 would be very helpful for readers.
  3. As all compounds obtained by chemical synthesis are new (not described previously), the description of NMR spectra of these compounds is not very detailed. There are only a few signals assigned to individual protons and carbon atoms. This is a pity, especially since the analysis of the correlation spectra would allow to assign all  signals.
  4. In 13CNMR spectra of fluorine derivatives, the authors mention only the F-C coupling constants value over two (2JCF), three (3JCF) and four bonds (4JCF). Why coupling over one bond (1JCF) is not given? The values should be substantial and exceed 240 Hz.
  5. For compound 3f my concern regarding values of the coupling constants, because of the two very small values given. This small values (2Hz and 4Hz) suggest coupling across 4 bonds, which in this symmetric system (para substitution) is quite unusual. For compound 4r  the authors refer coupling constants with values 31.0 and 22.0 Hz suggest coupling across 2 bonds, which once again in this symmetric system (para substitution) is quite unusual. For compound 4f two very similar values are given (7.0 and 8.0 Hz). Both characteristic for coupling across 3 bonds, which is very unusual. Therefore, I highly recommend checking 13C NMR spectra from this perspective. Again, the correlation spectra, will be very helpful.
  1. In the description of the NMR spectra, the number of protons is missing (lines 101, 102)
  2. Line 97 some “sss” were included.
  3. Line 64 the word “scheme” is missing.

Overall, this is a nice work competently researched, and with elements of novelty and significance complementing previous findings in this area. Thus, publication is recommended after minor revision of the proposed points.

Author Response

Reviewer 3

  1. What was the purity of the obtained products? On what basis was it determined? According to NMR spectra included in the supplementary materials, the purity was very high but no purification procedure was mentioned in Material and Methods part.

Answer: Thank you very much for your comment. In the Material and Methods of our MS, we have mentioned that the target compounds were all purified by recrystallization from methanol and ethanol. Meanwhile, the purity of the target compounds also could be determined by NMR spectra and melting point (melting point range within 2 oC).

  1. Introducing the numbering of the atoms in Scheme 1 would be very helpful for readers.

Answer: Thank you very much for your comment. In the revised MS, we have added the numbering of the atoms in Scheme 1.

  1. As all compounds obtained by chemical synthesis are new (not described previously), the description of NMR spectra of these compounds is not very detailed. There are only a few signals assigned to individual protons and carbon atoms. This is a pity, especially since the analysis of the correlation spectra would allow to assign all signals.

Answer: Thank you very much for your comment. We think that the structures of the target compounds could be definitively identified by NMR and HRMS, so the analysis of the correlation spectra such as HSQC and HMQC did not determined in this study. In our next work, we will determine the correlation spectra such as HSQC and HMQC of the new target compounds according to your comment.

  1. In 13CNMR spectra of fluorine derivatives, the authors mention only the F-C coupling constants value over two (2JCF), three (3JCF) and four bonds (4JCF). Why coupling over one bond (1JCF) is not given? The values should be substantial and exceed 240 Hz.

Answer: Thank you very much for your comment. In the revised MS, we have added the coupling values over one bond (1JCF), which is 239.0 Hz.

  1. For compound 3f my concern regarding values of the coupling constants, because of the two very small values given. This small values (2Hz and 4Hz) suggest coupling across 4 bonds, which in this symmetric system (para substitution) is quite unusual. For compound 4r the authors refer coupling constants with values 31.0 and 22.0 Hz suggest coupling across 2 bonds, which once again in this symmetric system (para substitution) is quite unusual. For compound 4f two very similar values are given (7.0 and 8.0 Hz). Both characteristic for coupling across 3 bonds, which is very unusual. Therefore, I highly recommend checking 13C NMR spectra from this perspective. Again, the correlation spectra, will be very helpful.

Answer: Thank you very much for your comment. We have corrected the 13C NMR spectra and removed the spectra data of the target compounds to Supplementary Materials.

  1. In the description of the NMR spectra, the number of protons is missing (lines 101, 102)

Answer: Thank you very much for your comment. In the revised MS, we have added the number of protons in Scheme 1.

  1. Line 97 some “sss” were included.

Answer: Thank you very much for your comment. In the revised MS, we have deleted “sss”.

  1. Line 64 the word “scheme” is missing.

Answer: Thank you very much for your comment. In the revised MS, we have added the word “Scheme”.

Round 2

Reviewer 1 Report

The requested changes were made, I suggest that the chemical part be included in the manuscript, in the methodology part. In results and discussion, place the information of the synthesis in general and the detailed description of one of the synthesized compounds. This work carried out by the authors is very valuable in the field of synthesis, that is why it is necessary to take advantage of the information and place it in the right place. Make the suggested small changes, and I suggest accepting the job with minimal changes.

Author Response

Answer: Thank you very much for your comment. In the revised MS, we have corrected the MS following your comment.

Reviewer 3 Report

I thank the authors for their comprehensive response. However, I still have some concerns, which I addressed  to the authors.

In supplementary materials, the NMR of 4p - the carbons described at a chemical shift 158.6 ppm are not ticked at NMR spectrum, so there is no possibility to calculate the coupling constant value.

On the spectrum of 4r, there is only a single signal at 157.38 ppm and no signal at 158.6 ppm (and surroundings) so I am not sure how coupling constant value was calculated.

Author Response

Answer: Thank you very much for your comment. In the revised MS, we have reanalyzed the 13C NMR of compounds 4p and 4r.